# Indicators, Goals, and Assessment of the Water Sustainability in China: A Provincial and City—Level Study

**DOI:** 10.3390/ijerph20032431

**Published:** 2023-01-30

**Authors:** Peipei Zhang, Yuanyuan Qu, Ye Qiang, Yang Xiao, Chengjun Chu, Changbo Qin

**Affiliations:** 1Institute of Strategic Planning, Chinese Academy of Environmental Planning, Beijing 100043, China; 2Yantai Consulting & Designing Institute of Environmental Engineering, Yantai 264000, China; 3The Center for Beautiful China, Chinese Academy of Environmental Planning, Beijing 100043, China; 4Center of Environmental Status and Plan Assessment, Chinese Academy of Environmental Planning, Beijing 100043, China

**Keywords:** water sustainability, indicators, 2035 goals, provincial evaluation

## Abstract

The United Nations and scholars called for more attention and efforts for cleaner water and water sustainability. This study established a water sustainability evaluating method framework, including indicators, goals, and methods and performs provincial and city−level assessments as case studies. The framework involves six fields, surface water quality, marine environmental quality, water−soil−agriculture, water infrastructure, water conservation, aquatic ecology, water−efficient use, and pollutant emission reduction. The methods innovatively integrate multi fields and concerns of water sustainability while providing a goal−oriented evaluation and implementing the United Nations’ call for the refinement and clarification of SDGs. China’s overall water sustainability was evaluated as 0.821 in 2021, and have performed well in surface water quality, sea quality, water conservation, and aquatic ecology fields while performing poorly in the water−soil−agriculture field. The overall strategy, policy, and action for water sustainability could be developed based on the evaluation. The water sustainability evaluation presented the regional and field/indicator differentiations. It is necessary to implement regionally classified policies and differentiated management for sustainable water development. The correlation analysis with socioeconomic factors implies the complicated and intimate interaction between socioeconomic development and water sustainability while revealing that development stages and the inherent conditions of natural ecology and water sources bring about the differentiations. A comprehensive evaluation of water sustainability may be three−dimensional, involving water quality and ecology, development related to water, and water resources and utilization.

## 1. Introduction

The United Nations called for more attention and efforts for cleaner water and water sustainability in the Sustainable Development Goals (SDGs) Report 2022 [1]. Water sustainability is the continual supply of clean water for human uses and for other living things [2]. In China, a populous country encountering water resource scarcity and spatially uneven distribution [3,4,5], achieving water sustainability would be more urgent [6,7]. During the past decades, though China has significantly improved its water environment by reducing pollution and water eutrophication [8,9], rivers and lakes that are still unsafe for human use account for a considerable proportion [10]. From 2015 to 2021, the proportion of China’s surface waters meeting national Grade Ⅲ increased from 64.5% to 87.0%, while that not meeting national Grade Ⅴ decreased from 8.8% to 0.9%. However, China’s per capita water resources are only 2098.5 m^3^, equivalent to less than 50% of the world’s overall. In recent years, China has changed its water strategy from pollution control to collaborative governance of water resources, water ecology, and water environment, reflecting the strategic thinking upgrading from simple pollution control to water sustainable management [11,12]. However, there are still ongoing challenges. In China’s “Fourteenth Five−Year Planning” for river basin management and water security, water resource conservation, water pollution reduction, and water ecological protection and restoration are still separated. A pressing concern is that the evaluating and managing frameworks focusing on water sustainability have not been established, especially since the indicators and goals are not determined and applied. Moreover, basin water environment management has received more academic attention, but comprehensive human−water sustainability study based on human activity units was relatively lacking [13,14,15,16,17,18,19]. Provinces (including municipalities and autonomous regions, the same as below) and cities are the two basic administrative units for environmental governance and water sustainability management in China. A systematic evaluation method integrating the water sustainability indicators and goals while covering province− and city−level would be of both research and practical value.

Goal 6 of SDGs requires cleaner water and sanitation, while goals 11, 14, and 15 propose water sustainability from the perspective of the human community and water life. Previous studies considered water security and water management as crucial aspects of water sustainability [20,21,22,23,24]. In more specific studies, water quality, water recycling, wetland protection, etc., are also considered [25,26,27,28]. The correlation and relationship between water resources−security−environment and agriculture, energy, and other fields have also been studied by previous research [29,30,31,32,33,34,35]. Referring to China’s SDGs localization studies, we systematically identified the relevant fields and indicators of water sustainability. Specifically, a systematic study proposed a classification–coordination–collaboration framework of sustainable development, highlighting the synergy between SDGs and policies, economy, science and technology, culture, and so on [36]. In the comprehensive sustainability assessment study, water resources, stability, and pollution have been the primary concerns [37]. In the sustainable development progress over China’s space and time, generally based on the SDGs framework, water stress and treatment have been included [38]. In an environmental sustainability study at the national and provincial levels, water sources, water ecology, and water pollution have been considered [39]. In a study that assessed the sustainable development goals achieving progress in 15 countries, including China, water utilizing intensity is considered primarily for water sustainability [40]. Apparently, the water quality, food−water productivity, and terrestrial ecology have not been highlighted, which should be reinforced in the water sustainability evaluation indicator framework. Water environment quality is an intermediary and characterizing variable between human development and natural resources, but it has not been reflected appropriately in previous water sustainability assessments and studies. Especially for China and other developing countries, the quality of the water environment is a non−ignorable aspect of water sustainability [41,42,43]. Moreover, specific and quantifiable goals for water sustainability indicators could significantly improve the guidance and reliability of evaluation but have rarely been achieved in the SDGs localization studies. Previous studies have commonly used standardized and normalized evaluation methods to assess water sustainability rather than provide exact water sustainability goals. This evaluation method is not conducive to proposing reasonable water sustainability goals and ignores the possible innate conditions of water sustainability among the assessment samples.

This study aims to establish a water sustainability evaluating method framework, including indicator, goal, and evaluating methods, which could achieve the quantitative and goal−oriented assessment of water sustainability in Chinese provinces and cities. The methods innovatively integrate multi fields, and concerns of water sustainability, especially water quality and water−related production, have drawn less attention in previous studies. The goal−oriented assessment could present more instructional and practical results. Specifically, we first identify and determine the indicators for water sustainability. The indicator framework generally covers all water−related fields of SDGs and has been expanded based on the characteristics and needs of China’s water sustainability. Second, we study and propose the goal for each indicator. China puts forward a goal of achieving a beautiful China by 2035, which is considered a pathway of SDGs localization [44,45,46]. To make the long−term goals more reliable and match China’s national strategic vision, we set the targeted year of the evaluation indicators as 2035. Third, we evaluated the water sustainability for China’s 31 provinces and 336 prefectural and above cities using the indicator−goal framework and multi−evaluating methods. The framework and evaluation results could provide methodological guidance and practical application reference for water sustainability assessment at different levels.

## 2. Materials and Methods

### 2.1. Framework, Indicators, and Goals

This study focuses on healthy water ecology and the environment while the safe supply of water resources for water sustainability. We establish the water sustainability evaluation framework from six fields, including surface water quality, marine environmental quality, water−soil−agriculture, water infrastructure, water conservation, aquatic ecology, water−efficient use, and pollutant emission reduction. The roadmap is provided in Appendix A. A total of 32 indicators, some of which include several sub−indicators, were proposed for water sustainability assessment. We determined the 2035 goals for every indicator, referring to several information sources. The first is the latest specified goals in policies, plans, and action programs issued by Chinese officials. The second is summarizing and analyzing mid− and long−term goals proposed by China’s local governments and competent departments. The third is to use good performance of indicators in specific provinces or refer to other countries. Figure 1 and Table 1 presents the water sustainability evaluation indicators and 2035 goals. Detailed information for indicators and goal determination is provided in Appendix A.

### 2.2. Water Sustainability Performance Evaluation Method

The water sustainability performance evaluation is based on the actual value and goals by 2035 for the indicators. By referring to the methods in authoritative research, such as the OECD distance measure [47] and the Progress measure based on Eurostat’s report [48], we use the single factor indicator as the basic method for the water sustainability evaluation. The equation is as follows, for the positive attribute indicator in Table 1, the water sustainability performance is evaluated as follows:(1)$$WS_{i}=\left\{ \begin{array}{l} C_{i}\geq G_{i},\;1\\ C_{i}<G_{i},\;C_{i}/G_{i} \end{array}\right.$$
for negative attribute indicator, the water sustainability performance is evaluated as:(2)$$WS_{i}=\;\;\left\{ \begin{array}{l} G_{i}\neq 0,\;\left\{ \begin{array}{l} C_{i}\leq G_{i},\;1\\ C_{i}\in (G_{i},\;2G_{i}),\;2-C_{i}/G_{i}\\ C_{i}\geq 2G_{i},\;0 \end{array}\right.\\ G_{i}=0,\;\left\{ \begin{array}{l} C_{i}=0,\;1\\ C_{i}\neq 0,\;1-C_{i}/100\% \end{array}\right.\; \end{array}\right.$$
where *WS_i_* is the water sustainability performance for the indicator *i*; *C_i_* is the actual value of indicator *i*; *G_i_* is the 2035 goal for the indicator *i*. The *WS* results are dimensionless, while the value is between 0 and 1. At the provincial level, we use the last available data to evaluate water sustainability. The 2021 data are used for most indicators, while the 2020 data are for a few. The arithmetic means of in−field indicators are calculated to characterize the water sustainability of the fields, and the field mean value reflects the overall water sustainability.

In 2015, the Chinese government issued the *Water Pollution and Control Action Plan* and intensified the action for water pollution prevention and control since then [8,49]. Therefore, we perform a developing progress assessment between 2015 and 2021 for parts of the water sustainability indicators to complement the evaluation from a time−varying perspective. Considering the cross−time comparability of data, we re−screen the water sustainability indicators and exclude indicators that have not changed during 2015–2021 due to policy settings. The five excluding indicators are the proportion of groundwater not meeting national Grade V, wetland protection rate, proportion of ecological protection red−line areas to land area, continental natural coastline retention rate, and proportion of marine ecological protection red−line areas to the jurisdictional sea area. Between 2015 and 2021, the water sustainability development for the individual indicator is evaluated by dimensionless changing rate. The water sustainability development for fields is calculated using the indicator average. For all field results, those less than −1 and greater than 2 were assigned values of −1 and 2, respectively.

### 2.3. Water Sustainability Evaluation at the City Level

Considering the data availability, we performed city−level water sustainability based on a selected version of the indicator framework. The city−level water sustainability evaluation framework involves three fields: water quality, water infrastructure, and water conservation and rational utilization. The indicators are presented in Table 2. Due to the limited applicability of 2035 goals to cities, we use the best−performance method to assess city−level water sustainability. Specifically, we set the actual value of the 15th percentile in all evaluated cities as the performance goal for individual indicators. Subsequent evaluation methods and equations are the same as at the provincial level. We include 336 of China’s 337 prefecture−level and above cities in the city−level water sustainability evaluation, including 4 municipalities, 292 prefecture−level cities, 7 regions, 30 autonomous prefectures, and 3 leagues. Sansha in Hainan province is excluded because data were not available.

## 3. Results and Discussion

### 3.1. National Water Sustainability

From a national perspective, China’s overall water sustainability was evaluated as 0.821 in 2021. The field water sustainability of surface water quality, sea quality, water conservation, and aquatic ecology were relatively high, scoring 0.961, 0.937, and 0.930, respectively. On the contrary, the water−soil−agriculture field scored only 0.344, revealing the conflict between water, land, soil and primary industry production in China, which constrained the water sustainability from the producing sector. Specifically, the high application intensity of pesticides and fertilizer and the share of fish harvesting has severely damaged the agricultural− and aquaculture−sustainability of waters. Moreover, though the indicator performance of groundwater quality is relatively acceptable in the field of water−soil−agriculture, it still has a large gap compared with surface water and marine environmental quality indicators. Therefore, to comprehensively improve water sustainability, China should first strengthen the shortcomings in the field of water−soil−agriculture. The water infrastructure and water efficient use and pollutant emission reduction fields scored 0.877 and 0.876 in 2021 water sustainability, respectively. Water supply, sewage drainage, and garbage harmless treatment facilities in towns and rural areas still need continuous improvement.

From a developing perspective, the performance of most water sustainability indicators has been improved between 2015 and 2021, except for the proportion of hydropower, implying comprehensive and significant progress in China’s water sustainability. The surface water and sea quality have improved significantly with field water sustainability development scores of 0.310 and 0.541. Combined with the two fields’ water sustainability evaluation results, China has made considerable efforts and achieved remarkable results in enhancing surface water and seawater quality in 2015–2021, also proving that China’s water sustainable development focused on quality improvement in this period. In addition, improving water infrastructure construction and promoting water−related green development (the 2015–2021 water sustainability development scored 0.324 and 0.303, respectively) have also contributed to better water sustainability in China. By contrast, the water−soil−agriculture and water conservation, and aquatic ecology fields improved relatively insignificantly, with water sustainability development scores of 0.174 and 0.039, respectively. The performance improvement of water ecology and water conservation indicators is a long−term process that can be understood and accepted. The improvement in water−related green production was insignificant, which requires attention and improvement.

### 3.2. Provincial Water Sustainability

Figure 2A provides the field−specific results of water sustainability of 31 Chinese provinces. Provinces in southeast and southwest China generally scored higher in water sustainability evaluation, while the Beijing−Tianjin−Hebei and surrounding areas showed poor water sustainability. Most provinces performed well in the surface water quality field except for a few, such as Tianjin. Provinces with relatively poor surface water quality are generally located in northern China, especially the Beijing−Tianjin−Hebei and surrounding areas (also involving Shanxi, Shandong, and Henan) and northwest provinces (including Inner Mongolia, Shaanxi, Gansu, Qinghai, Ningxia, and Xinjiang), which is inferred to be directly related to the scarcity of water resources there. In Jilin, Heilongjiang, and Yunnan, higher pollution loads and background values could be the principal cause of poor surface water quality. Seawater quality showed a relatively significant gap with the surface water in coastal provinces, indicating stage differences in water quality improvement. Therefore, the continuous improvement of marine environmental quality can be determined as a more dominant indicator in China’s water sustainability improvement strategy. For the water−soil−agriculture field, the central and western regions, especially inland provinces, performed significantly better than the eastern and coastal provinces, revealing the intrinsic linkage of socioeconomic development and land use intensity with water sustainability in this field. Therefore, differentiated strategies should be implemented to enhance water−soil−agriculture sustainability in these two regions. Specifically, the eastern and coastal provinces should jointly promote water−soil ecological environment restoration and reduction in ecological and environmental load of production. The central−western and inland provinces should focus on organizing production and living activities based on the water−soil ecological and environmental capacity, especially considering the fragility and diversity of the ecology and environment there [50,51]. Some provinces showed water infrastructure weakness, most of which were the western provinces with vast land and sparse population, such as Inner Mongolia, Heilongjiang, Tibet, and Qinghai. On the other hand, water infrastructure shortcomings in populous provinces such as Shanxi, Liaoning, and Henan should draw relatively urgent attention. We can find apparent disparities in the provincial performance of fields water conservation and aquatic ecology and water efficiency use and pollutant emission reduction. For the water conservation and aquatic ecology field, the Beijing−Tianjin−Hebei and surrounding areas, Yangtze River Delta (including Shanghai, Jiangsu, and Zhejiang), and northwest provinces performed relatively poor due to the contradiction between high−intensity economic development and limited ecological resources. For the water efficient use and pollutant emission reduction field, the Beijing−Tianjin−Hebei and surrounding areas and part of the northwest provinces, which are not rich in water resources, urgently need to improve.

Figure 2B presents the water sustainability development between 2015 and 2021 for 31 Chines provinces. All the provinces showed an overall improvement in water sustainability, though that inter−province varied widely. The water sustainability development scores in 15 provinces were higher than 0.5, showing significant improvement, while all field−specific water sustainability has increased in 18 provinces. From the water sustainability and the development scatters (Figure 3), we could find high water sustainability and high development cluster, including Chongqing, Sichuan, Guizhou, Yunnan, Shaanxi, and Qinghai, which are all western economically developing provinces. However, the good performance of water sustainability development in these provinces was dominated by water infrastructure, and even the surface water quality scores in Guizhou and Shaanxi have decreased. Meanwhile, surface water quality in most economically relatively undeveloped provinces such as Hainan, Guizhou, Tibet, and Gansu has deteriorated. Contrary, the improvement of water sustainability in economically relatively developed provinces has been dominantly contributed by green development and quality improvement. This revealed the phrased differences in the development of water sustainability among Chinese provinces, and the continuous improvement of water sustainability would be driven by infrastructure construction, quality improvement, water resource−ecology restoration, and then comprehensive water sustainability gradually.

### 3.3. Water Sustainability of Cities

Figure 4 and Figure 5 present the city−level water sustainability evaluation results. Water sustainability was evaluated as higher than 0.6 in only 63 cities (accounting for 18.75%) while lower than 0.4 in 116 (34.52%). The average and median values for the 336 cities’ water sustainability were 0.470 and 0.440, respectively. From a city perspective, we can observe significant regional differentiation in water sustainability. The poor water sustainability performance cities were majorly distributed in Beijing−Tianjin−Hebei and surrounding areas, other economically developed regions such as the Yangtze River Delta and Pearl River Delta (including nine cities of Guangzhou, Shenzhen, Zhuhai, etc.), and the northeast regions. The water quality field generally performed poorer than the other two fields, with a city−average score of 0.416 and a city−median score of 0.369. The city−average and city−median values were 0.514 and 0.523 for the water infrastructure field, while those for the water conservation and rational utilization field were 0.479 and 0.457, respectively. Considering that the city−level water sustainability evaluation is based on the 15th percentile best performance of internal data, the city distribution of evaluation results demonstrated the differences in water sustainability between cities to a certain degree. Moreover, the general opposite trends of the high−low scores distribution between water infrastructure and other fields also demonstrated the developing and regional differences in the drivers of water sustainability improvement discussed in Section 3.2. From 2015 to 2021, water sustainability in 243 cities has improved, accounting for 72.32% of the total, and the water sustainability for water quality, water infrastructure, and water conservation and rational utilization fields increased in 229 (68.15%), 282 (83.93%), and 122 (36.31%) cities. Widespread improvements in water quality and water infrastructure have been the dominant drivers of water sustainability improvements. In contrast, the increased water use intensity and water ecological pressure posed constraints on water sustainability.

We performed a correlation analysis using the city−level water sustainability evaluation results as dependent variables to understand the linkages between water sustainability and socioeconomic development. After running a 1−sample Kolmogorov−Smirnov (K−S) test and obtaining non−normal distribution, we chose Spearman’s rank correlation [52,53]. The selected independent variables include the per capita GDP, residential population, and per capita water source, and the results are shown in Table 3. We can observe that per capita GDP and residential population were negatively correlated with water sustainability. Nevertheless, the negative correlation was not very significant, implying the inter−city differentiation of the correlations. Similar results could also be found between water quality and the two independent variables, while water conservation and rational utilization field showed a relatively negative correlation. Indicatively, socioeconomic development showed a roughly negative impact on water sustainability, especially on water quality and water conservation, and rational utilization. In contrast, water sustainability showed a significantly positive correlation with per capita water resources, revealing that abundant water resource is the most non−ignorable factor for maintaining and improving water sustainability. There is seemingly an impossible triangle between water resource environment−economic and social development−water infrastructure, but this is a phrased characteristic of overall water sustainability, that is, a contradiction between development and water sustainability. With further economic and social development and improvement of water infrastructure, water ecology and environment will be further improved and restored. As these three dimensions achieve stability and harmony, so will comprehensive water sustainability.

### 3.4. Discussion and Policy Recommendations

This study performs a systematic evaluation of water sustainability in China at the national, provincial, and city levels. The framework established in this study involves more fields than previous works, especially considering water quality and marine environment provide a reference for China and other developing countries to assess water sustainability intuitively and effectively. The water−soil−agriculture and water−related ecological protection fields are also valuable expansions and demonstrations of the water sustainability assessment system. The goal−oriented evaluation method can provide a more quantifiable assessment and a clearer vision of water sustainability simultaneously, implementing the United Nations’ call for the refinement and clarification of SDGs, which would be of high practical value. Based on the methodology and results of this study, goal−oriented policies and strategies for water sustainability can be developed to enhance the implementation of SDGs in China. The results provide further inspiration for water sustainability policies. From the national evaluation, we recommend performing water sustainability evaluation regularly and institutionally while adjusting the overall water sustainable development strategy according to the fields’ performance. Moreover, the goals for water sustainability should be studied and determined more deeply. The provincial and city−level water sustainability evaluation results highlight the regional and field/indicator differentiations, which are considered correlated with development stages and the inherent conditions of natural ecology and water sources. For countries or regions with large spatial differences in water ecological and environmental endowments, such as China, it is necessary to implement classified policies and differentiated management for sustainable water development. Strategies such as regional water sustainable development appraisal, ecological protection compensation, and water resource finance can be considered. The correlation analysis implies the complicated and intimate interaction between socioeconomic development and water sustainability. Economic development could play a positive role in water sustainability from the policy and action levels, and the evaluation and governance of water sustainability force green and high−quality economic development. Our framework includes some indicators reflecting the efforts to protect the ecology and promote water sustainability, which could be a manifestation of this recommendation. We also suggest performing water sustainability assessments by basins and administrative units simultaneously, which may provide information and guides with more policy−making and action−organizing value. For instance, strategies such as water sustainability compensation, crucial field identification for water sustainability, and the region− and basin pathways for synergistic improving water sustainability can be designed and implemented based on this. A period− and regional−classified correlation analysis based on multi−year data could provide a more detailed decomposition of the development−sustainability relationship. Based on the correlation analysis, we have discovered a triangularly impossible relationship that exists in stages between the water resource environment−economic and social development−water infrastructure. With further thinking, a comprehensive evaluation of water sustainability may be three−dimensional, involving water quality and ecology, development related to water, and water resources and utilization.

### 3.5. Limitations

Though this study has referred to informative and insightful works by previous scholars and performed the evaluation using relatively rich data from national, provincial, and city levels, some methodological limitations still should be addressed. First, due to the indicator uncertainties and data unavailability, some field and indicators have not been considered in the water sustainability evaluation framework, such as the state of water ecology, aquatic biodiversity, emerging pollutants in water bodies, and the impact of shipping on water sustainability. A more comprehensive framework and indicator system will be of higher academic and practical value. Second and also limited by data, the city−level and provincial evaluation could not use the same framework, limiting the comprehensiveness of the city−level evaluation and preventing more in−depth analysis and the proposal of more specific water sustainability policy recommendations. Third, the 2035 goals use uniform rather than province− or regional−specific values, inevitably leading to insufficient fairness for some indicators’ evaluation, especially for the indicators reflecting the inherent conditions of water ecology and water resource. Though the goal−oriented evaluation has some advantages, as we mentioned in Section 3.4, the goals could considerably determine the evaluation results for the indicators, so the inappropriateness of goals may significantly affect the evaluation results.

## 4. Conclusions

This study establishes a water sustainability evaluation framework focusing on the application at the administrative unit level and performs provincial and city−level evaluation case studies. The field− and indicator−specific characteristics of water sustainability are also discussed. The results prove that the established framework and indicator−goal evaluation system could support quantitative and informative analysis of water sustainability at various levels. Especially for developing countries and territories like China, the indicators and goals system could support a water sustainability stage identification and strategy guide.

## Figures and Tables

**Figure 1 ijerph-20-02431-f001:**
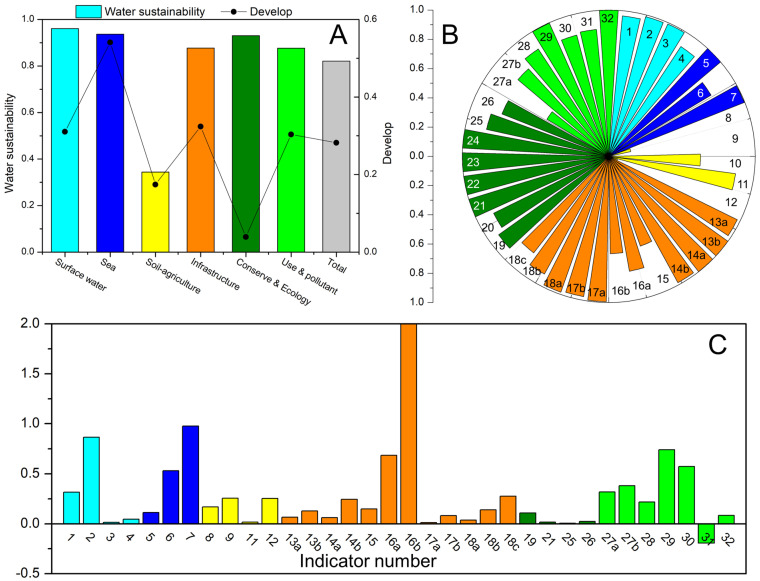
The field—(**A**) and indicator—specific (**B**) results of China’s water sustainability evaluation and the field—((**A**) the scatter—line) and indicator–specific (**C**) development of water sustainability between 2015 and 2021 (the number is the same as in Table 1).

**Figure 2 ijerph-20-02431-f002:**
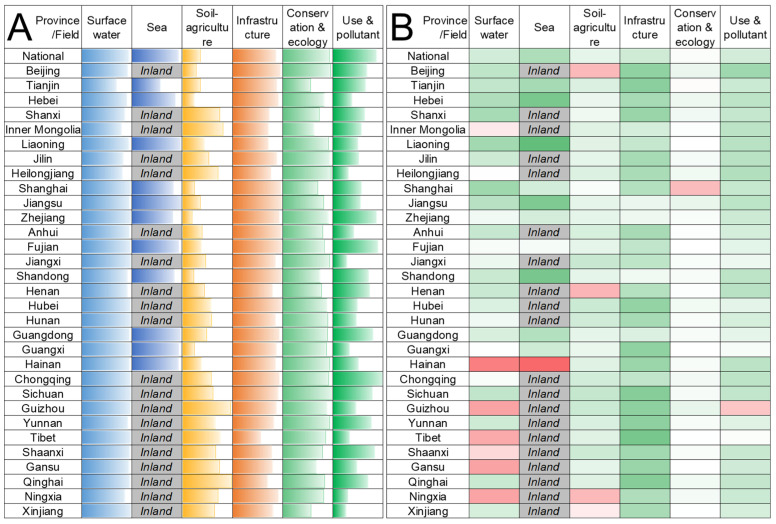
Provincial water sustainability evaluation (**A**) and development (**B**) between 2015 and 2021.

**Figure 3 ijerph-20-02431-f003:**
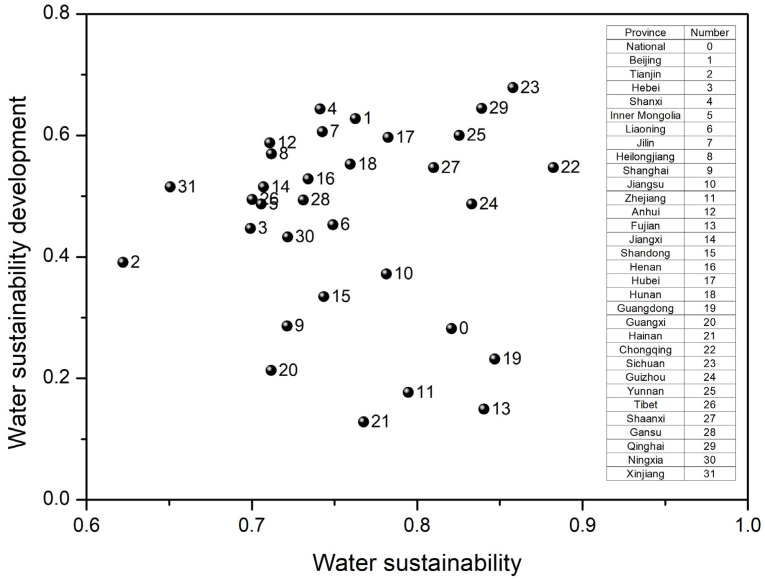
Water sustainability and development scatters for provinces.

**Figure 4 ijerph-20-02431-f004:**
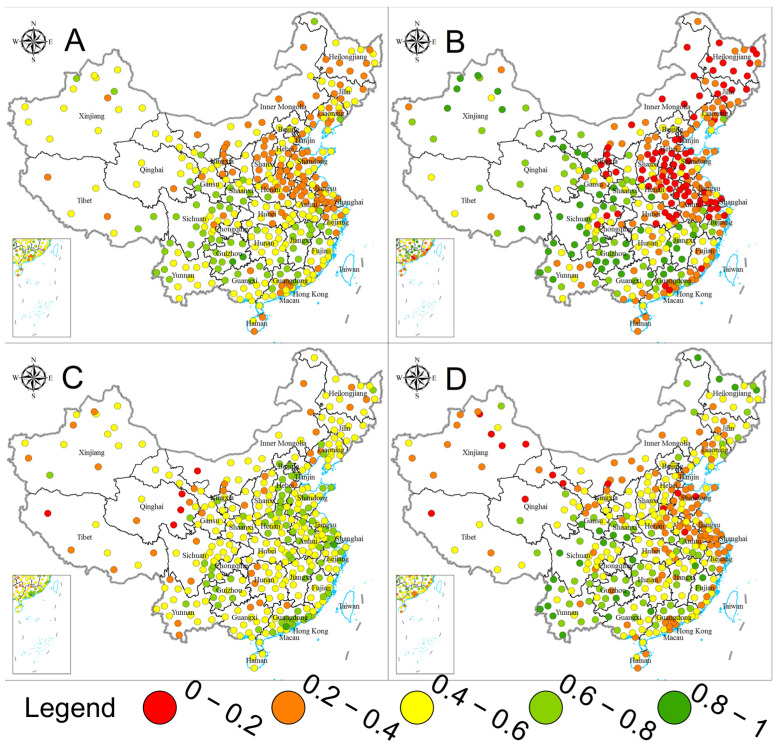
Water sustainability by city (**A**) for water sustainability, (**B**) for water quality, (**C**) for water infrastructure, and (**D** for water conservation and rational utilization).

**Figure 5 ijerph-20-02431-f005:**
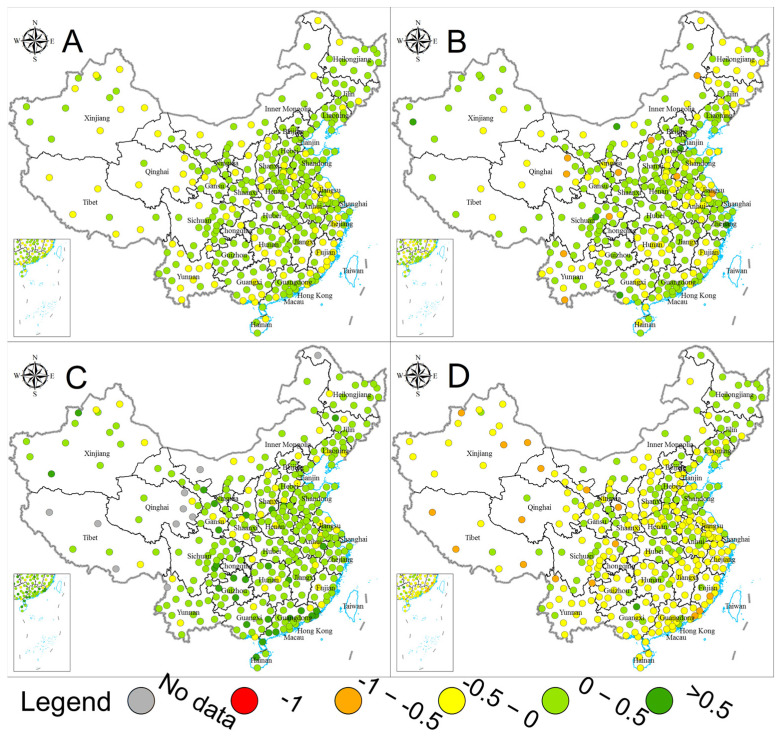
Water sustainability development between 2015 and 2021 by city ((**A**) for water sustainability, (**B**) for water quality, (**C**) for water infrastructure, and (**D**) for water conservation and rational utilization).

**Table 1 ijerph-20-02431-t001:** The water sustainability evaluation indicator−goal framework.

Field	No.	Indicator (Unit, Attribute)	2035 Goal
Surface water quality	1	Proportion of surface waters meeting national Grade III (%, +)	88
	2	Proportion of surface waters not meeting national Grade V (%, −)	0
	3	Standard compliance rate of centralized drinking water sources (%, +)	95
	4	Standard compliance rate of water function areas (%, +)	99
Marine environmental quality	5	Proportion of seawaters meeting national Grade II (%, +)	80
	6	Proportion of sea−entering rivers meeting national Grade III (%, +)	88
	7	Proportion of sea−entering rivers not meeting national Grade V (%, −)	0
Water−soil− agriculture	8	Fertilizer (pure conversion) per unit cultivated land area (kg/hm^2^, −)	220
	9	Pesticide usage per unit cultivated land area (kg/hm^2^, −)	2.5
	10	Proportion of groundwater not meeting national Grade V (%, −)	15
	11	Effective utilization coefficient of irrigation water (/, +)	0.7
	12	Proportion of natural capture production of aquatic products (%, −)	10
Water infrastructure	13	Urban/suburban household sewage treatment rate (%, +)	100/95
	14	Harmless treatment rate of urban/suburban household waste (%, +)	100/100
	15	Proportion of villages and towns that treat household sewage centralized (%, +)	90
	16	Harmless treatment rate of organized towns/townships household waste (%, +)	95/85
	17	Urban/suburban water supply (%, +)	100/100
	18	Centralized water supply of organized towns/townships/administrative villages (%, +)	100/100/100
Water conservation and aquatic ecology	19	Forest cover rate (%, +)	26
	20	Wetland protection rate (%, +)	60
	21	Proportion of nature protected areas to land area (%, +)	18
	22	Proportion of ecological protection red−line areas to land area (%, +)	25
	23	Continental natural coastline retention rate (%, +)	35
	24	Proportion of marine ecological protection red−line areas to jurisdictional sea area (%, +)	30
	25	Ecological Environment Index (/, +)	60
	26	Proportion of water and soil erosion area (%, −)	23
Water efficient use and pollutant emission reduction	27	Water consumption per unit of GDP/industrial added value (t/10^4^ Yuan, −)	36/25
	28	Chemical oxygen demand emissions per unit of GDP (kg/10^4^ Yuan, −)	Cumulative decrease 18% vs. 2020
	29	Ammonia nitrogen emissions per unit of GDP (kg/10^4^ Yuan, −)	
	30	Total Phosphorus emissions per unit of GDP (kg/10^4^ Yuan, −)	
	31	Proportion of hydropower (%, +)	18
	32	Proportion of water consumption to water resources (%, −)	23.33

Note: Indicators with (+) are positive indicators: the larger the indicator, the better the performance. Indicators with (−) represent a negative indicator: the larger the indicator data, the worse the performance. The GDP and added value are measured by constant 2020 price.

**Table 2 ijerph-20-02431-t002:** The water sustainability indicators for city−level evaluation.

Field	Indicator (Unit, Attribute)
Water quality	Environmental quality index of surface water (/, −)
	Environmental quality index of ground water (/, −)
	Environmental quality index of water sources (/, −)
	Proportion of seawaters meeting national Grade II (%, +)
Water infrastructure	Per capita household sewage discharge (kg/day)
	Per capita household waste generation (kg/day)
	Per capita treatment capacity of household sewage (kg/day)
	Per capita harmless treatment capacity of household waste (kg/day)
	Urban drainage network density (km/km^2^)
Water conservation and rational utilization	Forest cover rate (%, +)
	Proportion of nature protected areas to land area (%, +)
	Proportion of marine nature protected areas to land area (%, +)
	Ecological Environment Index (/, +)
	Proportion of water consumption to water resources (%, −)
	Per capita urban water consumption (t)

Note: Indicators with (+) are positive indicators: the larger the indicator, the better the performance. Indicators with (−) represent a negative indicator: the larger the indicator data, the worse the performance.

**Table 3 ijerph-20-02431-t003:** Spearman’s rank correlation fitting results.

Spearman’s Rank Correlation Coefficient	Per Capita GDP	Residential Population	Per Capita Water Resource
Water sustainability	−0.231	−0.201	0.535
—Water quality	−0.085	−0.280	0.446
—Water infrastructure	0.101	0.333	−0.298
—Water conservation and rational utilization	−0.464	−0.211	0.622

## Data Availability

Not applicable.

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
