# Peer review of "Indicators, Goals, and Assessment of the Water Sustainability in China: A Provincial and City—Level Study"

_ijerph, 2023, doi:10.3390/ijerph20032431_

Round 1

Reviewer 1 Report

This work establishes a framework for water sustainability and presents the provincial and city-level evaluations. The research content of this work is rich and logical. Therefore, my opinion about this paper is positive. I would like to mention, however, some aspects that should be considered before publication: 

1)The authors should first define the water sustainability more clearly.

2) In the introduction, the authors wrote that “specific and quantifiable goals for water sustainability…but have rarely been achieved in the SDGs localization studies”, please describe that in more details.

3) The method section should be re-organized for more readable. For example, the second paragraph of subsection 2.1 should not appear in the text but in appendix. Evaluation criteria and calculation methods of some Chinese localized indicators, such as III or V water should be explained in detail to let readers understand its specific meaning.

4) Some of the indicators in Table 1 (also Table 2) are so Chinese that it may be difficult for international readers to understand. I suggest an indicator interpretation in appendix.

5) Information presented in Figure 5 is generally repeated with Figure 3 and Figure 4. I suggest putting it in appendix. Similarly, Figure 1 and Figure 2 could be merged in one, and Figure 3 and Figure 4 could be scaled down, for the consideration of saving space and facilitating reading.

6)In the last paragraph of 3.1, “Water-related green production has been relatively slow,” is unclear. The authors might mean “the improvement for water-related green production was insignificant”.

7) The authors mentioned a word "staged" several times, and I think some of the expressions are inappropriate. I suggest checking relevant sentences and revising them.

8) The map-figures provided in manuscript do not include province names, which would confuse the readers when reading related text, it is recommended to add.

9) Discussion section is missed. If I understand appropriately is has been presented with results. Therefore, the Results should be Results and discussion, and the discussion sentences should follow the result and be better highlighted instead of jumbled together.

10) The city level assessment lost over a half indicators, which should be discussed as a limitation and uncertainty.

11)Correlation between water infrastructure and quality could be analyzed and discussed in more details.

12) How the methodology and results in this study support policy making or activities? Please specify.

13)The writing of the manuscript needs to be improved by a native English-speaking expert. Some spelling and grammar errors should be checked and revised, such as the indicators 2 and 7 "not meet" should be "not meeting", the "WSPi and WSNj" in line 144 without subscript, etc.

Author Response

Reviewer 1

This work establishes a framework for water sustainability and presents the provincial and city-level evaluations. The research content of this work is rich and logical. Therefore, my opinion about this paper is positive. I would like to mention, however, some aspects that should be considered before publication: 

1)The authors should first define the water sustainability more clearly.

Response: we added a definition of water sustainability in the first paragraph of Introduction and a framework definition at the beginning of Methods.

2) In the introduction, the authors wrote that “specific and quantifiable goals for water sustainability…but have rarely been achieved in the SDGs localization studies”, please describe that in more details.

Response: we pointed out the methodological and practical irrationality of the commonly used evaluation method for water-related sustainable development after this sentence.

3) The method section should be re-organized for more readable. For example, the second paragraph of subsection 2.1 should not appear in the text but in appendix. Evaluation criteria and calculation methods of some Chinese localized indicators, such as III or V water should be explained in detail to let readers understand its specific meaning.

Response: we added a Supplementary materials and moved the indicator interpretation and target determination in it.

4) Some of the indicators in Table 1 (also Table 2) are so Chinese that it may be difficult for international readers to understand. I suggest an indicator interpretation in appendix.

Response: referring to our response to comment 3), we added the indicator interpretation in appendix for the Chinese specific indicators.

5) Information presented in Figure 5 is generally repeated with Figure 3 and Figure 4. I suggest putting it in appendix. Similarly, Figure 1 and Figure 2 could be merged in one, and Figure 3 and Figure 4 could be scaled down, for the consideration of saving space and facilitating reading.

Response: we re-drew Figure 1 and combined the former Figure 1 and 2 in it. The former Figure 3 and 4 has also been combined in one while deleting the last column of them. Therefore, the overlapping contents in Figure 5 (now the Figure 3) is not of concern and could be retained in the manuscript.

6)In the last paragraph of 3.1, “Water-related green production has been relatively slow,” is unclear. The authors might mean “the improvement for water-related green production was insignificant”.

Response: agreed. We revised the description as the reviewer suggested.

7) The authors mentioned a word "staged" several times, and I think some of the expressions are inappropriate. I suggest checking relevant sentences and revising them.

Response: we checked the word in the manuscript and replaced it by "phased", "developing", or deleted it.

8) The map-figures provided in manuscript do not include province names, which would confuse the readers when reading related text, it is recommended to add.

Response: we added the province names in the current Figure 4 and 5 (the former Figure 6 and 7).

9) Discussion section is missed. If I understand appropriately is has been presented with results. Therefore, the Results should be Results and discussion, and the discussion sentences should follow the result and be better highlighted instead of jumbled together.

Response: we have revised the chapter title and made changes accordingly. Moreover, combining the suggestion of Reviewer 2, we added a subsection 3.4 Discussion and policy recommendations to perform the discussion more deeply.

10) The city level assessment lost over a half indicators, which should be discussed as a limitation and uncertainty.

Response: we added this limitation in current subsection 3.5.

11) Correlation between water infrastructure and quality could be analyzed and discussed in more details.

Response: we added analysis on water infrastructure and quality in subsection 3.3. Moreover, the relationship between socioeconomic development and water sustainability that reflecting by infrastructure-water quality correlations is also further discussed in the subsection 3.4.

12) How the methodology and results in this study support policy making or activities? Please specify.

Response: we added subsection 3.4 Discussion and policy recommendations. In subsection 3.4, we described how our methods, ideas, and multi-level evaluation support policy-making and provided recommendations.

13)The writing of the manuscript needs to be improved by a native English-speaking expert. Some spelling and grammar errors should be checked and revised, such as the indicators 2 and 7 "not meet" should be "not meeting", the "WSPi and WSNj" in line 144 without subscript, etc.

Response: we checked the English and revised the errors.

Reviewer 2 Report

Dear Authors, 

Consider below comments in revision version of your paper below:

1. The importance of your paper in both the abstract and introduction sections is not clear and also, and more evidence should be mentioned regarding the research gap. 

2. I strongly suggest that a number of official statistics related to the quality and quantity of water in China are mentioned in the introduction section.

3. It is suggested to mention a road map of the entire research process at the beginning of the materials and methods section because it is like a big picture.

4. Managerial Insides should be added to the research and the executive dimensions and implementation of the results of this research in the real world should be mentioned.

5. Discussion should be done deeply. You should evaluate your outcomes with critical analysis. For this goal, please find the link below:

https://www.uow.edu.au/student/learning-co-op/assessments/critical-analysis/

Author Response

Consider below comments in revision version of your paper below:

  1. The importance of your paper in both the abstract and introduction sections is not clear and also, and more evidence should be mentioned regarding the research gap. 

Response: we mentioned the importance of this study in abstract and the last paragraph of Introduction. We reorganized the second paragraph of Introduction and mentioned the research gap combining with the comment 2) of Reviewer 3.

  1. I strongly suggest that a number of official statistics related to the quality and quantity of water in China are mentioned in the introduction section.

Response: we added the official statistics related to China's water quality and the improvement in the introduction section.

  1. It is suggested to mention a road map of the entire research process at the beginning of the materials and methods section because it is like a big picture.

Response: we added a roadmap in Supplementary materials (Figure A1) and mentioned it at the beginning of the materials and methods section.

  1. Managerial Insides should be added to the research and the executive dimensions and implementation of the results of this research in the real world should be mentioned.

Response: combining the comment 12) of Reviewer 1, we added a subsection 3.4 Discussion and policy recommendations and specified and highlighted the policy recommendations in it.

  1. Discussion should be done deeply. You should evaluate your outcomes with critical analysis. For this goal, please find the link below:

https://www.uow.edu.au/student/learning-co-op/assessments/critical-analysis/

Response: we added a subsection 3.4 Discussion and policy recommendations. The outcome of this study is analyzed using critical analysis in current subsection 3.4 and 3.5. Specifically and according to the framework of critical analysis, the Description and Analysis is provided in subsection 3.4, the Evaluation is provided in subsection 3.4 and 3.5.

Reviewer 3 Report

I read carefully manuscript number:ijerph-2166609, the manuscript entitled: "Indicators, Goals, and Assessment of the Water Sustainability in China: A Provincial and city-level Study". Check the English Grammar. The English language is moderate. Please check all parts of the manuscript and correct grammatical errors. The authors should ask the help of native English speaking proofreader, because there are some linguistic mistakes that should be fixed. Nevertheless, the manuscript is not acceptable in its current form. I attached my reviewer comments in the PDF file. Authors should apply all of my comments.

Author Response

I read carefully manuscript number:ijerph-2166609, the manuscript entitled: "Indicators, Goals, and Assessment of the Water Sustainability in China: A Provincial and city-level Study". Check the English Grammar. The English language is moderate. Please check all parts of the manuscript and correct grammatical errors. The authors should ask the help of native English speaking proofreader, because there are some linguistic mistakes that should be fixed. Nevertheless, the manuscript is not acceptable in its current form. I attached my reviewer comments in the PDF file. Authors should apply all of my comments.

Response: we check the English and correct grammatical and spelling errors. Other comments have also been applied in this revision. Specific response are as follows.

  1. The abstract section need to complete with more information. The abstract should improve this section.

Response: combining with comment 1) of Reviewer 2, we revised the abstract and added more information, including the novelty and

  1. The literature review is too general and thus can’t indicate any novelty of the current study. It is better that explain more about the novelty of manuscript in introduction section. The manuscript has not quite innovative. Please explain about its novelty.

Response: we reorganized the second and third paragraph of Introduction and mentioned the novelty of this study in more details. Specifically, the previous studies on China's water-related sustainable development have been viewed in detail while several high-quality references are added. The novelty of water sustainability for the fields and evaluating methods have been elicited in the second by reviewing the shortcomings of previous studies in terms of water sustainability fields and evaluation methods. In the revised third paragraph of Introduction, we highlighted the novelty of our study framework and evaluation. Moreover, in the added subsection 3.4, the novelty has also been discussed as the Reviewer 2 suggested.

  1. The relevant reference may be of interest to the author according below:

Ostad-Ali-Askari, K. Review of the effects of the anthropogenic on the wetland environment. Appl Water Sci 12, 260 (2022). https://doi.org/10.1007/s13201-022-01767-4

Response: we cited this work as reference [27].

  1. Methodologies used in the manuscript should describe clearly. Tools for objective function optimization are unclear in the methodology.

Response: we improved the methodologies according to this reviewer while referring to Reviewer 2. Specifically, we first added a roadmap presenting the entire research process in Supplementary materials (Figure A1) and mentioned it at the beginning of the materials and methods section. Second, we moved the goal determination to Supplementary materials and added an indicator interpretation to make the description of indicators clear. Third, we added the sources and references for the indicator goals determination in Supplementary to make the tools for objective function optimization clear.

  1. Water sustainability performance evaluation method section was not written clearly. So, it must improve this section.

Response: in our view, the evaluation method for water sustainability is relatively clear. Only the Equation (1) was presented confusedly. Therefore, we split the former Equation (1) into current Equations (1) and (2) to make it clearer.

However, the calculation method for water sustainability development between 2015 and 2021 has not been written clearly. We re-wrote that in this revision. Specifically, we deleted the former Equations (2) and (3) and directly describe the calculation method of development in concise language.

  1. Result section was not written clearly. More explain about details of result.

Response: in our view, the result details have been presented enough. Considering this reviewer has not suggest what to "explain" clearly, we referred to the other reviewers' comments and added the discussion and policy recommendations to make the Results and discussion clearer. Please see the revised subsection 3.4 and 3.5 for details.

  1. Methodological limitations and future research recommendations section was not written clearly. So, it must improve this section.

Response: we moved the future research recommendations to newly added subsection 3.4 and reorganized the first paragraph of subsection 3.5 (the former 3.4). With this modification, subsection 3.5 becomes 3.5 Limitations and would be clearer.

  1. The conclusion section was not written marginal. So, it should improve this section.

Response: conclusion was revised to be marginal.

Round 2

Reviewer 2 Report

In my opinion, this paper is enhanced after revision process and it can be accepted.

Reviewer 3 Report

I read the revised manuscript number: ijerph-2166609, the revised manuscript entitled: "Indicators, Goals, and Assessment of the Water Sustainability in China: A Provincial and city-level Study". In my point of view, result of this kind of research could be interesting and useful for many applications specifically. All previous comments were applied. The authors applied all comments point by point and I confirm their revision. The added information is important and useful and led to improve the manuscript. I accept the revised manuscript in this present form. I concur; the final decision is accept for publication.